# Emerging and Future Targeted Therapies for Pediatric Acute Myeloid Leukemia: Targeting the Leukemia Stem Cells

**DOI:** 10.3390/biomedicines11123248

**Published:** 2023-12-07

**Authors:** Lindsey A. Murphy, Amanda C. Winters

**Affiliations:** 1Department of Pediatrics, City of Hope Comprehensive Cancer Center, Duarte, CA 91010, USA; lmurphy@coh.org; 2Department of Pediatrics, University of Colorado School of Medicine, Aurora, CO 80045, USA

**Keywords:** pediatric AML, immunotherapy, targeted therapy, leukemia stem cell (LSC)

## Abstract

Acute myeloid leukemia (AML) is a rare subtype of acute leukemia in the pediatric and adolescent population but causes disproportionate morbidity and mortality in this age group. Standard chemotherapeutic regimens for AML have changed very little in the past 3–4 decades, but the addition of targeted agents in recent years has led to improved survival in select subsets of patients as well as a better biological understanding of the disease. Currently, one key paradigm of bench-to-bedside practice in the context of adult AML is the focus on leukemia stem cell (LSC)-targeted therapies. Here, we review current and emerging immunotherapies and other targeted agents that are in clinical use for pediatric AML through the lens of what is known (and not known) about their LSC-targeting capability. Based on a growing understanding of pediatric LSC biology, we also briefly discuss potential future agents on the horizon.

## 1. Introduction

Acute myeloid leukemia (AML) is an aggressive hematologic malignancy that affects just over 4 per 100,000 adults and 7 per 1 million children in the United States each year [1,2]. In adults, AML incidence increases with advancing age [1], and the clonal expansion of immature myeloid-lineage blasts is thought to be secondary to the sequential accumulation of multiple mutations that collectively confer a growth/proliferation advantage over normal hematopoietic populations [3]. In children, the genetic drivers of AML are fewer in number and tend to be sufficient to generate an acute leukemia phenotype as stand-alone mutations, suggesting biological differences between adult and pediatric disease [4]. However, in both age groups, outcomes for AML remain suboptimal. Despite intensive treatment regimens including high-dose chemotherapy and at times allogeneic stem cell transplant (SCT), approximately 30–40% of children and young adults and over 80% of elderly adults will relapse [1,2]. Relapsed AML is a scenario where the prognosis is particularly poor and for which additional treatment options including novel agents are a relatively unmet need [5,6]. A better understanding of the pathophysiology and mechanistic drivers of the disease will help guide future therapeutic advances.

Leukemia stem cells (LSCs) have been implicated in the origination, chemo-resistance, and relapse of both adult and pediatric AML. There has been a general drive among translational researchers and clinicians to better characterize these cells to understand which aspects of their biology might be targeted for therapeutic purposes. The general hypothesis is that LSC-targeting agents, as an integral part of AML treatment, will improve event-free and overall survival for patients. While our knowledge of adult LSCs is substantial and growing constantly, research into pediatric LSCs is relatively nascent. The purpose of this review is to provide an overview of the pediatric LSC literature, within the framework of what is already known about adult LSCs, and identify which novel agents are thought to have the highest likelihood of successfully eradicating these leukemia-initiating cells.

## 2. Discovery and Evolving Knowledge of Stem Cells in Adult AML

### 2.1. Functional Definitions and Immunophenotype

The concept of a myeloid LSC first took shape in the mid-1960′s [7] and was further delineated in the 1980′s, when it was seen in clonogenic assays and murine engraftment models that there was a relatively rare population of AML cells that could recapitulate the phenotypic diversity of the disease [8,9]. Early studies of LSCs in adult AML showed the presence of functionally defined leukemia-initiating cells exclusively in the CD34+CD38- phenotypic compartment, irrespective of the patient immunophenotype, and correlated the frequency of this compartment in patient samples with survival outcomes [10,11]. More recent evaluations of the CD34+CD38- LSC phenotypic assumption have correlated the abundance of the CD34+CD38- compartment at diagnosis with post-treatment survival in patients having achieved remission and demonstrate that the quantitation of the abundance of LSC in this way can supplement a more conventional assessment of measurable residual disease (MRD) [12]. CD123 has also been demonstrated to be highly expressed on LSCs but, unlike CD34, generally has minimal or variable expression on hematopoietic stem cells (HSCs), raising the possibility of a therapeutic window for therapeutic targeting [13]. Incidentally, CD123 has also been shown to be a marker of stem cells in high-grade myelodysplastic syndrome (MDS), which carries a significant risk of transformation to AML [14].

Additional seminal studies have sought to more precisely quantify LSC frequency through limiting dilution experiments in immunodeficient mice; these studies highlighted the diversity of AML immunophenotypes in primary patient samples and revealed that LSC activity was not restricted to the CD34+CD38- cells, nor even to those lacking lineage markers [15]. Not only has interpatient heterogeneity been shown, but within individual patients, LSCs can have multiple immunophenotypes [16,17], and LSC surface marker expression even changes after chemotherapy in many cases [16]. Therefore, while many cell surface proteins have been proposed as LSC markers [13,17,18,19,20,21,22,23], their fidelity across the breadth of AML patients as well as through therapy for a single patient remains questionable.

### 2.2. LSC Transcriptional Signatures

An alternative approach to the identification and study of LSCs has been the development of LSC gene expression signatures. Eppert et al. published one of the first LSC gene signatures by comparing gene expression microarray data of sorted populations of functionally validated LSCs to populations of non-LSC blasts [24]. This group identified 42 genes whose expression was enriched in LSCs, confirmed that many of these genes were also expressed in normal HSCs, and correlated high expression levels of these genes with poor survival outcomes in three independent adult AML cohorts [24]. One of the most frequently cited LSC gene expression signatures is the LSC17 signature developed by Ng and colleagues in 2016 [25]. It was developed based on sorted cell populations from 87 adult AML patients that were functionally evaluated for the presence or absence of LSC activity. The most upregulated genes in the LSC+ populations compared to the LSC- populations underwent sparse regression analysis in order to identify the minimal gene list that was still prognostic of survival outcomes in a training cohort of adults with AML [25]. These 17 genes were further validated in three additional cohorts of patients, with high LSC17 scores associated with poor event-free and overall survival [25].

In 2015, investigators also published a DNA methylation signature based on functionally validated LSC and non-LSC populations from 15 adults with AML [26]. The majority (91%) of the over 3000 differentially methylated regions were hypomethylated in LSCs compared to non-LSCs, representing differential expression of 71 genes [26]. Only 6 of these 71 genes were described in the previous LSC gene expression signatures [26]. It was found that many *HOXA* cluster genes were upregulated in LSCs, irrespective of genetic driver mutations [26]. Expression of this LSC epigenetic signature was also correlated with outcomes in multiple patient cohorts [26]. Epigenetic dysregulation is a recurring theme in the adult LSC literature [27,28,29,30,31,32,33], with several potential therapeutic targets identified, including protein arginine methyltransferase (*PRMT6*) [28], vitamin D receptor (*VDR*) [31], and YTH N6-methyladenosine RNA binding protein F2 (*YTHDF2*) [33], to name a few.

### 2.3. LSC Metabolism, Microenvironment, and Drug Resistance

Additional characterization of LSCs in adult AML has involved the evaluation of metabolic and other functional vulnerabilities of these cells. One seemingly unifying feature of functionally validated myeloid LSCs is that they have lower levels of reactive oxygen species (ROS) compared to bulk blasts—a feature that has been used to isolate and study these cells irrespective of their immunophenotype [34,35]. In addition, LSCs in general have a greater dependence on oxidative phosphorylation (OXPHOS), with limited flexibility in transitioning to glycolysis or other energy sources, at least in chemotherapy-naïve cells [34,36]. In chemo-resistant LSCs, which are typically enriched at relapse, both amino acid metabolism [37,38,39] and fatty acid oxidation and transport [28,40,41,42,43] have been shown to play roles in greater metabolic resilience.

Interactions of LSCs with the surrounding niche have also been shown to play a role in their survival and propagation of AML. LSC-homing to the bone marrow is facilitated by CXCR4/CXCL12 interactions, which are promoted by AML-associated mutations such as *TET2* deletions through epigenetic modifications [27]. LSCs utilizing the CXCR4/CXCL12 axis to hide in the bone marrow have been shown to be more resistant to FLT3 inhibitors such as quizartinib [44]. Similarly, LSCs that had adapted to an adipose niche were shown to be resistant to conventional chemotherapeutic agents [42]. The microenvironment also plays a role in the immune evasion of LSCs [45].

Another feature of LSCs is their ability to directly eliminate chemotherapeutic agents through drug efflux pumps [46,47,48] or to upregulate autophagy in response to a variety of insults [49,50,51,52]. Autophagy inhibitors have shown preclinical utility in combination with epigenetic and metabolic agents [49,50] but may have antagonistic activity against certain kinase inhibitors [51].

As the breadth of LSC literature grows, additional options for LSC-targeting therapies are expected to come to light. Figure 1 provides a broad overview of LSC vulnerabilities, which have also been adeptly reviewed by others [53,54,55,56].

## 3. LSC Biology in Pediatric Myeloid Disease

With the upswell of interest and translational research focus on isolating and therapeutically targeting the myeloid LSC in adult medicine, it is not surprising that the literature on pediatric AML stem cells has also seen rapid growth in the past decade. Much of what has been published about pediatric AML stem cell biology borrows heavily from certain foundational tenets of adult AML stem cells, which can at times create limitations in the data. For example, many of the pediatric LSC studies base their investigation of pediatric LSCs on CD34+CD38- sorted populations of cells, which in most cases is enriching for LSCs in the context of adult AML but may miss the LSC heterogeneity that we now know is present [57,58,59,60,61,62]. With those caveats in mind, the findings from these studies could still prove to be therapeutically and prognostically interesting.

### 3.1. Immunophenotype

Potential immunophenotypic targets for pediatric LSCs have largely been explored in the context of comparing expression levels of published adult LSC markers in pediatric HSCs from healthy controls to CD34+CD38- populations from AML samples. Chavez-Gonzalez et al. demonstrated increased expression of both CD123 and CD96, but not CD117 or CD90, in putative LSCs versus normal HSCs [57]. Petersen et al. likewise confirmed increased expression of CD123, as well as CLL and IL1RAP, in primitive populations from AML patients versus healthy bone marrow donors, but noted that IL1RAP, CD93, and CD25 expression was not restricted to populations harboring AML-associated genetic mutations [58]. In retrospective analyses, both CD123 and CD200 expression have demonstrated a correlation with inferior survival and other poor prognostic features (high-risk genetics, persistent MRD) [60,63]. CD200 has been shown to play a role in immune evasion in both adult [64] and pediatric [65] AML, in part through decreasing STAT3-dependent cytokine secretion and reducing OXPHOS in T cells [64]. Another putative LSC marker, CD69, was identified as upregulated in LSCs from pediatric patients who failed to achieve remission with conventional chemotherapy [66].

### 3.2. LSC Transcriptional Signatures

The bulk of the existing investigation into pediatric LSCs has involved the validation of LSC-enriched gene expression signatures, building off of the adult literature. Duployez et al. initially evaluated the prognostic relevance of the LSC17 gene signature in two large retrospective pediatric AML data sets, finding that high expression of these 17 genes was strongly associated with inferior event-free and overall survival, including in multivariate analyses [67]. Elsayed et al. applied a least absolute shrinkage and selection operator (LASSO) Cox regression model to 32 of the 47 genes originally identified by Ng et al. and to survival data from the St. Jude AML02 trial and identified a “pLSC6” gene signature that was most discriminatory of survival outcomes in that patient cohort [68]. They then showed a superior association with the survival of pLSC6 compared to the LSC17 signature [68]. Most recently, Huang et al. reviewed all of the previously published LSC gene signatures in the context of RNA-seq data from 1503 diagnostic samples banked through four large Children’s Oncology Group (COG) trials [69]. This study showed that the larger LSC47 gene signature, from which the LSC17 signature was derived [25], was actually the most prognostic of outcomes within genetic risk groups compared to either LSC17 or pLSC6 [69]. The prognostic value of LSC47 was then validated in an independent cohort of 212 patients from St. Jude [69]. Together, the transcriptional data from large clinical cohorts provide corroborating evidence of the importance of LSC biology for childhood AML.

### 3.3. LSC Metabolism, Microenvironment, and Drug Resistance

Although metabolic and functional vulnerabilities of pediatric LSCs are relatively understudied, a growing body of literature is rapidly filling this knowledge gap. Alternative splicing events were identified in a small cohort of pediatric LSC samples that were subjected to single-cell proteogenomic analysis in parallel with normal HSCs [59]. Exon skipping was prominent in LSCs but not HSCs and included alternative splicing of *CD47*, which led to upregulation of this immune evasive molecule [59]. The exon skipping phenotype was also associated with increased transcript levels of *MCL1-L* and *BCL2-L* splice variants, both of which are pro-survival [59]. Rebecsinib, a splicing modulator compound, was shown to reduce viability and colony formation of pediatric LSCs but not cord blood HSCs [59], suggesting that this agent still in preclinical testing could have utility in treating pediatric AML. Rebecsinib has also been proposed as an LSC-targeting therapy in adult AML [70].

Another study compared single-cell RNA-seq data in paired pre- and post-chemotherapy bone marrow samples from 13 children with AML, using enrichment of chemo-resistant transcriptional signatures as a surrogate for LSC activity [66]. Traditional LSC signatures were seen in hematopoietic stem cell-like populations, while oxidative phosphorylation (OXPHOS) signatures were upregulated in progenitor populations, particularly from patients with more monocytic AML [66]. Gene set enrichment analysis also identified ribosome biology as upregulated in LSCs [66]. Furthermore, compared to LSCs from pre-chemotherapy samples, paired post-chemotherapy LSCs showed activation of genes responsive to reactive oxygen species and heme metabolism and maintained their LSC or OXPHOS gene signatures [66]. These metabolic features at baseline and in response to chemotherapy may inform the selection of adjunctive therapies for pediatric AML in the future. Similarly, analysis of large adult and pediatric transcriptional data sets showed that high RNA levels of the ATP-binding cassette (ABC) transport protein ABCA3 strongly correlated with leukemia-free survival as well as expression of the LSC17 gene signature [71], suggesting that LSCs, in particular, have enhanced drug resistance compared to bulk AML blasts and warrant careful consideration of their unique vulnerabilities to bypass this resistance.

With respect to the microenvironment, mouse studies suggest that inherent differences in niche crosstalk and stromal composition contribute to differences in self-renewal capacity and fitness between HSCs and LSCs, with neonatal marrow favoring normal HSC development and adult marrow favoring LSC propagation [72]. Introducing cells to the marrow of differently aged mice completely reversed the differences seen in phenotypes, suggesting that interactions with the bone marrow niche have significant implications for leukemogenesis [72]. Less is known about the contribution of the adipose niche in pediatric AML development and chemo-resistance, with conflicting data as to the relationship between body mass index (BMI) and outcomes in children. One meta-analysis of 11 articles including 2922 patients showed a significant correlation between higher BMI (≥ 85%) and both event-free and overall survival [73]. However, another multi-national study of 867 pediatric patients with newly diagnosed AML found no relationship between BMI and relapse risk, treatment-related mortality, or overall survival [74]. Additional preclinical and clinical studies are needed to better understand the contributions of different microenvironmental interactions toward pediatric LSC survival. Figure 2 summarizes what is known about pediatric LSC biology.

## 4. Immunotherapies

As preclinical and clinical knowledge around LSC immunophenotypes has grown, so has the interest in immune-based therapies. Immunotherapies differ significantly from chemotherapy in that they aim to stimulate a patient’s pre-existing immune system to preferentially target and kill the malignant cell population while minimizing harm to healthy cells. Cell membrane surface targets have been the main interest for the development of leukemia-directed immunotherapies, with a particular focus on antibody-based and cell-based therapies. While the landscape of acute lymphoblastic leukemia (ALL) therapy has changed completely as a result of immunotherapies, unfortunately, immunotherapy advances have not been as successful to date in the treatment of pediatric AML. When it comes to specifically eradicating the AML LSC population, an ideal target for an immunotherapeutic approach is highly and universally expressed on LSCs but absent in other normal cells and HSCs.

As previously discussed, research efforts to differentiate between the immunophenotypic profile of LSCs compared to HSCs are immense in order to identify and develop new immunotherapies that have an optimal therapeutic window. For AML, few targets have been identified that are exclusive to LSCs; many surface markers that are enriched on LSCs are also found on HSCs, thus the therapeutic window is relatively narrow. CD33 and CD123 are two of the most commonly pursued surface antigens as clinical targets but, unfortunately, many clinical trials for these antigens have seen unfavorable toxicity profiles. The off-target effects and prolonged myelosuppression that have been observed further emphasize the ongoing hurdles that need to be overcome in order to develop AML-selective and especially myeloid LSC- immunotherapies. What follows is a brief review of the immunotherapy agents that are recently or currently under investigation for pediatric AML and what is known about their LSC-targeting capabilities. Currently, active clinical trials that include pediatric patients are summarized in Table 1.

### 4.1. CD33

CD33 is a transmembrane receptor that is expressed highly on the majority of AML cells. It was identified early as a potential cell-surface therapeutic target, and its expression on AML LSCs exceeds that of HSCs [75,76]. Gemtuzumab ozogamicin (GO) is a CD33-targeting antibody-drug conjugate (ADC) consisting of a fully humanized CD33 antibody with a calicheamicin payload [77]. GO received Food and Drug Administration (FDA) approval in 2000 on the basis of early adult studies that demonstrated that GO could induce remissions in relapsed/refractory (R/R) AML [78]. However, due to toxicities including prolonged cytopenias and increased rates of veno-occlusive disease of the liver, as well as efficacy concerns in which the benefit of GO in R/R AML could not be confirmed in subsequent studies, GO was withdrawn from the market in 2010 [79]. These concerning toxicity profiles were likely due to higher GO dosing than what is currently used, as successive larger clinical trials confirmed demonstrable clinical benefits and more manageable toxicities with lower doses of GO [80,81,82,83,84,85]. Therefore, GO was reapproved by the FDA in 2016 for adults with relapsed and newly diagnosed AML and for children with relapsed AML [86].

The COG AAML0531 phase III clinical trial demonstrated that GO in combination with standard chemotherapy resulted in improved outcomes for certain subsets of pediatric patients with AML [81]. High CD33 surface expression was associated with inferior outcomes in children with de novo AML treated on AAML0531, and several studies have suggested that the benefit of GO is directly correlated with higher CD33 surface expression [81,87].

Recently, significant efforts have been made to optimize the potency of ADCs targeting CD33. Vadastuximab talirine is one such ADC that was found to be more effective at killing AML cells compared to GO but with significant hematologic toxicities described due to its small therapeutic window [88,89,90]. The phase III CASCADE study investigating this ADC in older patients with de novo AML was discontinued early due to substantial adverse events suggesting that normal HSCs as well as AML LSCs were being affected.

Lintuzumab-Ac225 is an alternative approach to CD33-targeting and consists of a radiolabeled anti-CD33 antibody. It delivers high-energy radiation over a short radius to CD33-expressing cells. Lintuzumab has been studied as a monotherapy and in combination with chemotherapy in adults with AML and has demonstrated significant antileukemic activity with high response rates, even in patients with high-risk and heavily pretreated disease [91]. More clinical trials of Lintuzumab are underway for adults but, it has not yet been tested in the pediatric population.

Bi-specific antibodies consist of two distinct antigen-binding domains to interact with two disparate cell surface proteins. Typically, these are designed to be able to recruit lymphoid cells to interact and kill malignant cells. Bi-specific T cell-engagers (BiTEs) have shown promising efficacy for the treatment of R/R ALL, and as a result of this success, there has been an increase in focused efforts to develop BiTEs for AML [92,93,94]. Specifically, these bi-specific antibodies for AML are constructed with a CD3 antibody moiety to employ T lymphocytes to recognize a cell surface target on the AML cells (e.g., CD33) causing T-cell mediated destruction. CD33 BiTEs have demonstrated effective LSC killing preclinically [95]. Several studies of AMG 330 have demonstrated safety and efficacy in the treatment of adult patients with AML, but this agent has not yet been investigated in children [93,95,96].

Chimeric antigen receptor (CAR) T cells have been extensively investigated with promising response rates in the context of R/R lymphoid malignancies. They are now being progressively pursued to treat several other types of malignancies. Unfortunately, in the setting of AML, CAR T cell therapies have not progressed as rapidly as they did for lymphoid malignancies, again due to difficulty in identifying target antigens that are restricted to the leukemia cells and LSCs.

CD33 has been one of the most studied antigen targets for AML CAR T cells. Preclinical research has demonstrated potent leukemia killing of CD33 specific CAR T cells and several clinical trials are ongoing investigating CD33 CAR T cells in both the adult and pediatric populations (NCT03927261, NCT03971799). To date, there are case reports of patients who have had promising responses to CD33 CAR T cells and have been successfully bridged to SCT [97]. Additionally, dual targeting CD33/CLL-1 CAR T cells are in clinical trials in China (NCT03795779, NCT05248685) in an effort to optimize efficacy and AML specificity of cytotoxic killing. One significant barrier to the success of CAR T cells in this context is that CAR T persistence is often associated with prolonged cytopenias without full hematopoietic recovery. Given the propensity for patients to become aplastic following these therapies, CD33 CAR T cells are largely being investigated as a bridge to SCT. This is in contrast to ADCs and BiTEs, which may be able to induce sustained remissions with less myelotoxicity. To combat the off-target effects seen with CD33-directed CAR T cells, research on the possibility of genetically inactivating CD33 in HSCs to permit CD33 CAR T cell-mediated cytotoxicity with increased specificity for leukemic blasts and LSCs is ongoing [98].

### 4.2. CD123

CD123 (the interleukin-3 [IL3] receptor alpha) was the first antigen that was identified to be specific to LSCs. Studies of mouse AML xenograft models established that the CD34+/CD38-/CD123+ population of AML cells held all of the detectable engraftment potential [13]. However, while CD123 is highly expressed on myeloid leukemic blasts in a majority of patients, it is also present on HSCs as well as more differentiated myeloid and lymphoid cells [99]. While the therapeutic window for CD123-directed therapies is narrow for this reason, interest in CD123 as a potential cell-surface therapeutic target persists [63]. Overall, clinical investigations of anti-CD123 immunotherapies have generally shown limited efficacy and unfavorable safety profiles due to the off-leukemia effects of CD123 targeting. As more trials have been developed and more patients treated with different CD123-directed therapies, it has been observed that patients with high CD123 benefit most from these immunotherapies [63,100].

While the IL3 signaling pathway is important for normal hematopoietic differentiation from HSCs, it has also been shown to be essential for LSCs [100]. In the efforts to develop agents that will be effective in killing AML LSCs, one approach has been to conjugate a cytotoxic drug to the IL3 ligand for the IL3 receptor CD123. Tagraxofusp is an IL3/diphtheria toxin fusion protein that is FDA-approved for the treatment of blastic plasmacytoid dendritic cell neoplasm (BPDCN), a rare malignancy that has high levels of CD123 expression [101]. In addition to normal HSCs, Tagraxofusp has shown potent cytotoxicity against AML LSCs in preclinical studies [102]. In initial trials in adults, clinically meaningful responses were observed after treatment with Tagraxofusp in a small subset of patients with R/R AML that received multiple lines of prior therapy, suggesting promising LSC-targeting [103]. Trials in both adult and pediatric patients with R/R AML are currently underway or in development to further evaluate the efficacy and toxicity profile of Tagraxofusp (NCT05476770, NCT05233618, NCT04342962, NCT05720988, NCT05716009).

IMGN632 is an ADC consisting of a CD123-targeting antibody and a DNA-alkylating payload, which, in preclinical studies, has demonstrated compelling antileukemic activity [104]. IMGN632 has been studied in early-phase clinical trials both as a single agent and in combination with chemotherapy in adults with AML with promising activity against AML with tolerable safety profiles (NCT04086264) [105,106]. Studies in adults are ongoing, but trials involving IMGN632 have not yet expanded to children.

Bi-specific engagers have also been investigated to target AML LSCs through CD123. These strategies include the CD123/CD3 dual-affinity retargeting protein (DART) flotetuzumab, the CD123-targeting bi-specific antibody XmAb14045, and the CD123/CD3 bi-specific IgG1 antibody JNJ-63709178. In early-phase clinical trials in adults with R/R AML, 30% of patients treated with flotetuzumab achieved responses, with the toxicity profile mostly including cytokine release syndrome (CRS) and infusion reactions [107,108]. Based on this data, a clinical trial is ongoing investigating flotetuzumab for the treatment of children with R/R AML (NCT04158739).

Given that significant CRS has been observed with CD123-targeted T cell engagers, the focus for cytotoxic antibody-directed therapies has shifted towards natural killer (NK) cells. SAR443579 is a trifunctional NK cell engager that targets CD123 while co-engaging NKp46 and CD16a on NK cells. In preclinical mouse and primate models, SAR443579 had potent antitumor effects through NK cell activation in the presence of AML blasts with a tolerable toxicity profile and minimal cytokine secretion [109]. In a first-in-human phase I/II trial of SAR443579 for adult patients with R/R AML, 3/23 total patients (13%) achieved a complete remission (CR), with the highest response rates at higher dose levels (3/8 patients, 37.5%) [110]. The most common adverse events were nausea and infusion reactions, with only two cases of low-grade CRS [110]. This first-in-human study of SAR443579 recently opened a pediatric arm enrolling patients greater than 1 year of age with R/R AML (NCT05086315).

Lastly, CAR T cells targeting CD123 are also in development with active CD123 CAR T cell trials enrolling adult and pediatric patients with R/R AML (NCT04265963, NCT04272125, NCT04318678, NCT04678336). Early reports from some of the pivotal CD123 CAR T cell trials have identified few significant off-leukemia effects, including minimal myeloablative toxicities, but with some myelosuppression and limited CAR persistence [111]. For these reasons, and similar to the approach for CD33 CAR T cells, CD123 CAR T cells are primarily being investigated as a bridge to SCT.

### 4.3. Checkpoint Inhibitors

Immune checkpoint antigen expression has been described in both bulk AML blasts and LSCs, and treatment concepts have been developed with the goal to block checkpoint molecules and thereby overcome immune evasion [112,113]. PD-L1 expression in AML LSCs is hypothesized to be due to (1) oncogenic mutations in key proteins that are part of the JAK-STAT and MYC pathway signaling cascades in LSCs and (2) cytokine production from AML cells leading to cytokine-induced expression of PD-L1 [114,115,116]. Additionally, certain drugs used in the treatment of AML, like hypomethylating agents, can promote increased expression of PD-L1 in AML blasts and LSCs [117]. Checkpoint inhibitors in combination with these chemotherapy medications have been evaluated in clinical trials for adult patients with AML. In a phase II study of azacitidine and nivolumab for R/R AML, 70 patients were treated with an overall response rate of 33% and overall survival of 19% [118]. A phase I/II study of the azacitidine/nivolumab combination is currently underway to evaluate safety and efficacy in children with R/R AML (NCT03825367). In a phase II clinical trial assessing the efficacy of nivolumab as a maintenance therapy for adults with high-risk AML in remission who were not being considered for SCT, 15 patients were enrolled, of whom 9 patients (60%) had detectable minimal residual disease (MRD) at time of enrollment [119]. The recurrence-free survival was 57.1%, and two of the nine patients with detectable MRD at the time of enrollment cleared their MRD while on treatment with nivolumab. This study showed only a modest effect on MRD eradication and survival when nivolumab was used as a single agent but has provided the basis for subsequent clinical trials investigating immune checkpoint blockade in combination with chemotherapy as a maintenance therapy in adults with high-risk AML.

### 4.4. CD47

CD47 is another immune checkpoint antigen that plays a role in how leukemia evades the immune system and has been found to be amply and invariably expressed on both AML LSCs and normal HSCs [120]. The CD47 antigen is also known as the ‘don’t eat me’ signal that mediates the escape of AML LSCs from phagocytosis by macrophages; blocking this signal can induce an immune response resulting in the killing of leukemic cells [121,122]. Magrolimab is an anti-CD47 antibody that induces tumor phagocytosis and eliminates LSCs in AML preclinical models through CD47 blockade. Magrolimab has also shown promising results in early-phase clinical trials in adults with newly diagnosed AML when combined with azacitidine, which has been shown to enhance CD47 expression [123]. In 16 evaluable AML patients treated with magrolimab and azacitidine on a phase Ib clinical trial, 11/16 (69%) had an objective response [124]. Additionally, CD34+CD38- identifiable LSC populations were measured by flow cytometry, and complete LSC elimination was observed in 10/16 (63%) of patients with myelodysplastic syndrome or AML that had a clinical response. This LSC eradication suggests the potential for durable responses in these patients, and clinical trials further investigating magrolimab efficacy are ongoing. Trials investigating magrolimab for the treatment of AML in the pediatric population have not yet been initiated.

### 4.5. Folate Receptor 1/Folate Receptor-Alpha

*CBFA2T3::GLIS2* translocations are associated with a particularly high-risk subtype of pediatric AML that is associated with megakaryoblastic phenotype in young non-Down syndrome patients [125]. This subtype of AML has proven to be highly chemo-refractory, and survival rates with traditional chemotherapeutic regimens and SCT are quite poor [125]. Transcriptional analysis of cells from patients with *CBFA2T3::GLIS2* AML has revealed high expression of the folate receptor-alpha gene (FOLR1), which correlates with prominent surface expression of the FOLR1 protein in the majority of patients [126]. Although no formal validation of FOLR1 as an LSC marker has been undertaken, preclinical modeling of STRO-002, an antibody-drug conjugate (ADC) against FOLR1 [126] and, separately, a FOLR1-specific chimeric antigen receptor T cell (CAR-T) [127], showed promising anti-tumor efficacy. Limited clinical experience with the use of STRO-002 for pediatric AML has been presented in an abstract form in which 16 total patients were treated with this ADC alone or in combination with other chemotherapeutic agents, with 7 (44%) achieving complete remission (CR) [128]. Updated clinical data are expected to be shared at the American Society of Hematology 2023 annual meeting.

An alternative FOLR1-targeting agent, ELU001, has also demonstrated preclinical efficacy in MV4-11 cells engineered to overexpress FOLR1 [129]. ELU001 is a C’Dot drug conjugate (CDC), a nanoparticle-based drug delivery system with 21 exatecan molecules per nanoparticle as a payload. A phase I clinical trial of ELU001 for children with R/R *CBFA2T3::GLIS2* AML is expected to open to enrollment in 2024 (NCT05622591).

## 5. Other Novel Agents

As has been noted above, immunophenotype is not the only or even the most consistent feature of myeloid LSCs in the pediatric or adult literature. Stem cell transcriptional, signaling, and metabolic programs have proven to be targetable features of LSCs as well. Figure 1 and Figure 2 highlight a few of the relevant aspects of LSC biology that have been explored for therapeutic intent. Below and in Table 2, we summarize non-immunotherapeutic agents that have been or will soon be in clinical trials for pediatric AML and their relevance to the concept of LSC eradication.

### 5.1. FLT3 Inhibitors

Fms-like tyrosine kinase 3 (FLT3), also known as CD135, is a receptor tyrosine kinase that is expressed on HSCs and has functions in normal hematopoiesis [130]. Activating mutations in FLT3 such as the internal tandem duplication (ITD) or tyrosine kinase domain (TKD) mutations are common in both adult and pediatric AML [4,131]; FLT3 ITD mutations, in particular, confer a poor prognosis [131]. Tyrosine kinase inhibitors (TKIs) that target FLT3 have generally improved survival both alone and in combination with conventional chemotherapy in adult clinical trials [132,133,134,135,136,137,138]. Safety and preliminary efficacy of FLT3 inhibitors quizartinib and sorafenib have also been demonstrated in pediatric early-phase trials [139,140]. In the COG trial AAML1031, the addition of sorafenib for FLT3-mutant AML prolonged event-free and disease-free survival and lowered relapse risk compared to historical cohorts of patients with FLT3-mutant AML (COG AAML0531) [141]. An ongoing phase III trial is evaluating the addition of gilteritinib to chemotherapy for all patients with FLT3 ITD or TKD mutations, including a maintenance phase of gilteritinib monotherapy for patients proceeding to SCT. Whether FLT3 is an important survival mechanism for LSCs is an open question. While multiple studies (adult and pediatric) have demonstrated the presence of FLT3 mutations in the phenotypic LSC compartment [142,143], it is notable that FLT3 mutations generally are considered a “late” stage modification in the evolution of adult AML [144], and approximately 50% of patients who relapse or progress on FLT3 inhibitor therapy lose their ITD mutations [145]. In one preclinical study, the FLT3 inhibitor sorafenib alone was insufficient to block LSC functionality in serial transplantation experiments of leukemia into mice, but the addition of all-trans-retinoic acid (ATRA) to sorafenib abrogated transmission of leukemia to secondary recipients [146], suggesting that there are combination therapies that can effectively eradicate LSCs. In addition to ongoing pediatric trials evaluating established FLT3 inhibitors, there are two phase I trials of newer agents pexidartinib [147] and MRX-2843 [148], both of which are active against common TKI resistance mutations such as the “gatekeeper” F691 mutation.

### 5.2. Menin Inhibitors

Translocations in the *KMT2A* gene, a master hematopoietic regulator, are associated with both acute lymphoblastic and myeloid leukemias and generally confer poor prognosis [4,131,149]. The LSC in *KMT2A*-rearranged leukemias is transformed by the *KMT2A* translocation in isolation and generally arises from a granulocyte-macrophage progenitor (GMP) rather than from the HSC [150]. *KMT2A* rearrangements activate a transcriptional program characterized by upregulation of *HOXA* genes and *MEIS1*, which are associated with lineage-inappropriate expression of stemness markers [149,150]. The N-terminus of KMT2A, which is conserved in leukemogenic fusion proteins, recruits transcription elongation complexes to target genes in a manner that is dependent on its interacting protein, menin [151,152]. VTP50469 was the first small-molecule inhibitor of the menin-KMT2A interaction; treatment of *KMT2A*-rearranged leukemia cell lines and primary samples with this agent led to downregulation of the classic *KMT2A* gene signature and prolonged survival (even cures) in mice [153]. Similar gene expression signatures and similar benefits from menin inhibition have been shown in preclinical studies for *NPM1*-mutant AML [154,155], *NUP98*-rearranged AML [156], and *UBTF* mutated AML [157,158]. Revumenib has been the first menin inhibitor in clinical trials for AML; the first-in-human phase I monotherapy results were recently published. Of 60 evaluable patients with R/R *KMT2A*-rearranged or *NPM1*-mutant AML, 18 (30%) achieved a composite complete remission (CRc), with complete cytogenetic remission in 64% of patients with *KMT2A*-rearranged AML who had clearance of bone marrow blasts [159]. Preclinical support for the utility of another menin inhibitor, ziftomenib, has been published for both adult and pediatric AML [160,161]. In the adult study, percentages of phenotypically defined stem/progenitor cells were reduced after ziftomenib treatment [160]. Preliminary data from KOMET-001, the phase I/II trial of ziftomenib in adult R/R AML, were presented in abstract form in 2022. At the phase Ib dosing of 600 mg, CRc was 33%, with 75% of these patients achieving MRD negativity; the overall response rate (ORR) was 42% for the entire cohort and 75% in patients experiencing differentiation syndrome [162]. Additional combination therapy trials of ziftomenib and revumenib are ongoing in the adult age group. For pediatric patients, ziftomenib is currently available only through expanded access, but revumenib and another compound, JNJ-75276617, are open or soon-to-be-open in combination therapy trials (Table 2).

### 5.3. Venetoclax

Venetoclax is a BCL-2 inhibitor that was designed as a BH3 mimetic compound to block interaction between BCL-2 and pro-apoptotic proteins [163]. While initially showing clinical promise in chronic lymphocytic leukemia (CLL) [164], venetoclax in combination with low-intensity therapies has revolutionized the care of elderly patients with AML, many of whom do not have intensive therapy options and are not medically fit for SCT [165,166]. It has also shown promise in younger adults in combination with both high- and low-intensity chemotherapeutic regimens [167,168]. BCL-2 has been shown to be expressed at high levels in canonical myeloid LSCs and to be associated with a dependence on OXPHOS [34]. Although there is literature to suggest that the canonical pro-apoptotic role of venetoclax is its primary mechanism [169], there is also a wealth of metabolic data that suggest that mitochondrial respiration in LSCs is affected and plays at least some role in the efficacy of venetoclax [36]. Both pathophysiologic hypotheses assume that venetoclax acts as a priming agent for AML cell death and so is most effective when combined with other agents. Resistance to venetoclax has been associated with metabolic switching of LSCs to alternative energy sources such as amino acid or fatty acid metabolism [39,40,43]. Phenotypically, this has been associated in adult AML with the presence of non-canonical monocytic LSC biology [35,170].

Multiple retrospective single-center studies [171,172,173,174] and one multi-center study [175] have evaluated the efficacy of venetoclax combinations in R/R pediatric AML, with ORR ranging from 12.5% to 75% and a significant number of patients reported as able to bridge to SCT. Venetoclax in combination with daratumumab and myeloablative preparatory regimens led to an 85% MRD-negative CR rate in pediatric patients proceeding to transplant with active disease and resulted in a 2-year EFS of 44% and OS of 65% [176]. A phase I trial combining venetoclax with high-dose cytarabine and idarubicin identified the RP2D as 360 mg/m^2^ and reported the total cohort ORR as 69%, with 70% CRc at the RP2D [177]. Although little is known about LSC-targeting capability or metabolic effects of venetoclax in pediatric AML, a recent single-cell analysis including five pediatric primary specimens confirmed that four out of five pediatric AML specimens had venetoclax-resistant subpopulations with unique metabolic features [178]. This suggests that at least some of the trends seen in adult AML may hold true in pediatric disease as well. Based on accumulating data about venetoclax resistance mechanisms, new rational combination therapies are being explored in preclinical [179] as well as adult clinical trial settings. Multiple combination regimens are currently in early-phase pediatric trials as well (Table 2).

### 5.4. PARP Inhibitors

Poly (ADP-ribose) polymerases (PARPs) are enzymes that catalyze the addition of polymerized ADP-ribose onto substrates, thereby modifying their function or stability [180,181]. They are primarily implicated in DNA damage response pathways [180,181]. PARP inhibitors such as olaparib or talazoparib combined with DNA damaging agents such as anthracyclines or topoisomerase inhibitors tend to show synergy because of impaired DNA repair in tumor cells [181]. This is the premise of the current phase I pediatric trial of talazoparib combined with topotecan and gemcitabine (POE22-01), which is not known to be LSC-targeting. However, in one adult study of LSCs, the lack of cell surface expression of natural killer group 2 member D (NKG2D) ligands was noted to be a consistent feature of these cells and mediated escape of immune surveillance by NK cells [20]. PARP inhibitors were found to induce NKG2D ligand expression on LSCs and enhance immune clearance [20]. Therefore, future trials may explore this mechanism in pediatric AML to ascertain the potential synergy between PARP inhibitors and immunotherapies, for example.

### 5.5. Epigenetic Modifiers

Given that altered methylation is a key feature of adult LSCs [26], it is logical to anticipate that epigenetic modifiers such as DNA methyltransferase inhibitors (e.g., decitabine), hypomethylating agents (decitabine or azacitidine), or histone deacetylase inhibitors (e.g., vorinostat) might add a benefit to combination regimens against pediatric AML. To that end, a phase I feasibility study of epigenetic priming with decitabine followed by standard induction chemotherapy in newly diagnosed pediatric patients with AML was conducted. It enrolled 25 patients and showed no difference in CR rates, MRD negativity rates, or DFS between decitabine priming and standard-of-care arms, although it was not powered for efficacy endpoints [182]. Methylation changes were seen in the decitabine arm, which is consistent with pharmacodynamic goals [182]. A larger phase I dose expansion trial of decitabine and vorinostat in combination with FLAG chemotherapy was conducted through the TACL (Therapeutic Advances in Childhood Leukemia & Lymphoma) consortium. This trial enrolled 37 patients with a median age of 8.4 years who had R/R AML [183]. The regimen was shown to be safe with no dose-limiting toxicities at the decitabine RP2D of 10 mg/m^2^ [183]. The ORR in 35 evaluable patients was 54% of which 90% were MRD-negative and 84% were able to be bridged to SCT [183]. Two-year OS was 75.6% for patients who became MRD-negative with therapy versus 17.9% for those who remained MRD-positive after treatment [183]. Real-world data from six centers, including 28 patients who were treated off-study but with the same regimen, showed similar safety and efficacy with an ORR of 63% and 93% of responders able to proceed to SCT [184]. Given the relative success of this regimen in a heavily pretreated population and the promise of venetoclax combination regimens, there is now an open trial adding venetoclax to azacitidine, vorinostat, and FLAG (NCT05317403) in hopes of demonstrating safety and enhanced efficacy.

### 5.6. Selinexor

Selective inhibitors of nuclear export (SINEs) are compounds that inhibit the protein exportin-1 (XPO1) [185], which has numerous ramifications in the context of cancer. Tumor suppressors are expected to be retained in the nucleus and thereby re-activated [185]. Mutant *NPM1*, a commonly mutated protein in adult AML, requires cytoplasmic localization for its leukemogenic activity, which is inhibited by treatment with SINE compounds such as selinexor or its more potent second-generation relative, eltanexor [186]. Selinexor has been shown to potentiate DNA damage induced by both cytotoxic chemotherapy [187] and venetoclax [188]. Furthermore, through the reduction of MCL1 protein levels in treated cells, eltanexor further potentiates the antileukemic effects of venetoclax [189]. Preclinical studies have shown that both selinexor and to a greater degree eltanexor target LSCs, as demonstrated by limiting dilution engraftment experiments in mice [190,191]. While selinexor monotherapy led to only a 14% response rate in R/R adult AML [192], combination trials with chemotherapy have shown more favorable response rates of 42–50% [185,193,194], aside from one study in elderly AML that showed worse outcomes with the addition of selinexor to standard chemotherapy [195]. A phase I trial combining selinexor with fludarabine and cytarabine in 18 pediatric patients with R/R AML demonstrated XPO1 target inhibition and led to an ORR of 47% [196]. Another phase I trial of these agents plus the addition of venetoclax is ongoing at multiple U.S. sites (Table 2).

### 5.7. Niclosamide

Niclosamide is an anti-parasitic agent that shows antineoplastic activity through inhibition of NFkB and cAMP-response element binding protein (CREB) as well as through upregulation of reactive oxygen species (ROS) [197,198]. A cell line CRISPR/Cas9 library screen identified mitochondrial metabolic pathways as significantly altered after exposure of AML cells to niclosamide—in particular OXPHOS, glycolysis, and mitochondrial membrane potential—suggesting effects on pathways essential to LSCs [199]. Indeed, preclinical studies showed decreased viability of CD34+CD38- cells and decreased colony-forming potential in AML primary samples, both of which suggest a selective reduction in the LSC compartment after niclosamide treatment [197]. There are minimal to no clinical data on niclosamide in AML; however, a phase I monotherapy trial is actively recruiting for pediatric patients with R/R AML (Table 2).

### 5.8. Uproleselan

Interaction of leukemia cells, including LSCs, with the microenvironment is thought to be protective against chemotherapy toxicity by a variety of mechanisms. LSCs in adults have been shown to express E-selectin ligands, which enhance binding to the bone marrow niche [200]. The E-selectin antagonist uproleselan showed preclinical efficacy in sensitizing AML cells to standard chemotherapeutic agents [200], and a recent phase I/II study in adults with R/R AML and elderly patients with newly diagnosed (ND) AML (phase II only) was recently published [201]. A total of 91 patients were enrolled, approximately 2/3 of which were R/R to prior therapies. Overall response rates were 41% in the R/R cohort and 72% in the ND cohort, with over half of responders achieving MRD negativity. Responses were correlated with E-selectin ligand expression on bulk blasts and on immunophenotypically defined LSCs (CD34+CD38-CD123+) [201]. Based on these data, the COG Cellular Therapies for AML Task Force is conducting a phase I/II trial of uproleselan combined with myeloablative preparatory regimens for allogeneic SCT in pediatric patients with AML (NCT05569512). The primary endpoint is to identify the RP2D of uproleselan with standard transplant conditioning regimens, and secondary endpoints include 12-month leukemia-free survival (LFS) and 2-year CIR and OS.

### 5.9. Enasidenib

Mutations in the isocitrate dehydrogenase-2 (*IDH2*) gene occur in 10–15% of adult AML [202] but only 2% of pediatric AML [203]. Mutations in *IDH2* affect cytoplasmic and mitochondrial metabolism and have also been shown to have epigenetic effects leading to differentiation arrest [204]. A selective small-molecule inhibitor of mutant *IDH2*, enasidenib, has been developed and tested in adult AML in both R/R and newly diagnosed patients. In the R/R setting, response rates for enasidenib monotherapy average 40% but with a response duration of only 5.8 months [205,206]. In propensity-matched cohorts, a survival benefit of ~33% was cited over conventional care (low-dose chemotherapy or supportive management) [207]. Response rates in the up-front setting in combination with azacitidine [208] or intensive chemotherapy [209] were higher—74% for enasidenib + azacitidine versus 36% for azacitidine alone and 63% for enasidenib + conventional chemotherapy. The drug appears to induce the differentiation of leukemic blasts into neutrophils in responding patients [202]. The role of LSC-targeting in the mechanism of action of enasidenib remains inadequately characterized.

### 5.10. Pevonedistat

The NEDD8-activating enzyme regulates the degradation of proteins involved in essential processes such as cell cycle progression and DNA damage repair [210]. A novel NEDD8-activating enzyme inhibitor, MLN4924 or pevonedistat, was developed to target cancer cells that rely heavily on protein turnover for these cellular processes. In vitro analysis of pevonedistat demonstrated inhibition of NFkB, increased ROS generation [210], and depletion of intracellular nucleotide pools contributing to increased DNA damage [211]. Co-treatment with pevonedistat and cytarabine in vitro led to increased incorporation of cytarabine into DNA synthesis and decreased colony formation of AML cell lines and primary samples [211]. The LSC-targeting capability of pevonedistat has not been well-characterized. However, clinical trials involving pevonedistat have not had the hoped-for success. Dose-limiting toxicities, particularly hepatotoxicity and multi-organ failure, are common [212]. Phase II and III studies randomizing between pevonedistat and azacitidine versus azacitidine alone have not demonstrated significant benefit to the addition of pevonedistat [213,214]. A phase I/II study of pevonedistat in combination with venetoclax and azacitidine showed similar CRc rates to venetoclax + azacitidine alone [215]. Early data from the phase I pediatric trial of pevonedistat, azacitidine, fludarabine, and cytarabine again suggest high rates of dose-limiting toxicities (25%), with only 3 of 12 patients achieving CR with incomplete count recovery [216]. Data maturation will be needed to determine whether this agent moves forward in later-phase pediatric trials.

## 6. Future Targets

### 6.1. Immunotherapies

Although not a comprehensive list, below are some potential immunotherapy targets for pediatric myeloid LSCs that may see future clinical development:CD70 is a tumor necrosis factor receptor ligand that is not normally expressed in normal tissues or on HSCs during hematopoiesis. It is upregulated on immune cells upon activation but not on resting B or T lymphocytes [217]. It has been demonstrated that CD34+ AML cells and LSCs express CD70 and its receptor CD27, that CD70/CD27 signaling in AML cells activates stem cell expression programs, and that the promoter for CD70 is sensitive to methylation [218,219]. For these reasons, blocking CD70/CD27 signaling in conjunction with hypomethylating agents is being considered as a potential treatment concept for AML. Currently, a CD70-targeting antibody, cusatuzumab, in combination with azaciditine or venetoclax, remains under clinical investigation with promising initial responses but short follow-up of treated patients to date [220].Surface expression of CD69 was enriched on LSCs from patients whose disease proved chemoresistant in one study, and CD69 expression in transcriptional data from large retrospective cohorts of pediatric patients correlated with poor outcomes [66]. Therefore, CD69 could represent a future LSC-targeting strategy for pediatric AML, although CD69 expression on regulatory T cells and other specialized T cell subsets may indicate unwanted side effects of immune dysregulation with CD69-targeting [221].CD200, because of its high expression on pediatric LSCs [60] and its association with immune evasion [65], may also be a future immunotherapy target.

### 6.2. Other Novel Agents

Although not a comprehensive list, below are some potential novel non-immunotherapy targets for pediatric myeloid LSCs that may see future clinical development:A 2021 study evaluated AML transcriptional data from 284 pediatric patients and found that high expression of calcitonin receptor-like receptor (CALCRL), a G-protein coupled receptor with roles in proliferation, apoptosis, and inflammation, was associated with inferior 5-year EFS and OS compared to those with low expression of CALCRL [222]. Antibody- or small molecule-based targeting of the CALCRL ligand calcitonin gene-related peptide (CGRP) are currently being investigated for migraine (NCT03432286, NCT05217927) and may warrant investigation as novel agents for both adult and pediatric AML. As noted above, given the association of LSCs with alternative splicing and exon skipping [59], a splicing modulator such as rebecsinib might be of future clinical interest in pediatric AML, particularly combined with anti-CD47 agents and/or pro-apoptotic agents such as venetoclax.Telomerase activity has been postulated to be a dependency in adult LSCs, and genetic deletion of an RNA template subunit (*TERC*) in mouse models of leukemia significantly impaired LSC functionality [223]. While children with AML tend to have lower telomerase activity on average than adults, those with higher telomerase activity had worse outcomes in a retrospective analysis [224]. Pre-treatment of six pediatric patient-derived xenograft (PDX) samples with the telomerase inhibitor imetelstat, alone or in combination with azacytidine- or cytarabine-based chemotherapy, reduced LSC viability, prolonged survival in primary murine recipients, and reduced engraftment into secondary recipients [225]. These data suggest that imetelstat could be an LSC-targeting therapy that should be prioritized for combination regimens. There is currently an ongoing clinical trial for adult MDS and AML that is enrolling participants at multiple Australian sites (NCT05583552).

There is substantial evidence in adult AML and mounting evidence in pediatric AML that these diseases arise as a result of a leukemia-initiating population of LSCs. Identification of phenotypic, metabolic, and other vulnerabilities of these cells is the most direct path forward toward the development of more effective and less toxic therapies for patients of all ages with AML. Prioritization of immunotherapeutic and other novel agents for which there is strong preclinical and/or adult clinical data for LSC-targeting should be standard practice for pediatric clinical trial design.

## Figures and Tables

**Figure 1 biomedicines-11-03248-f001:**
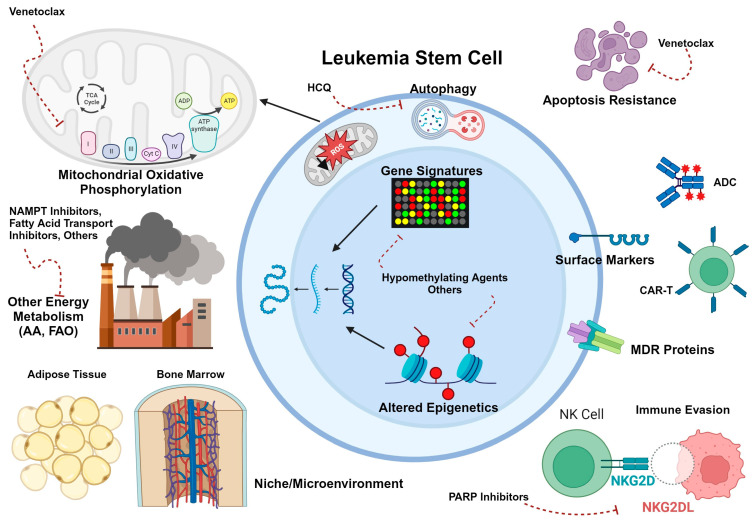
Schematic diagram of LSC biology in adult AML. Potentially targetable aspects are highlighted. These include LSC-selective immunophenotypes that can be targeted with antibody-drug conjugates (ADCs) or chimeric antigen receptor T cells (CAR-T); the crucial role of oxidative phosphorylation in metabolism; the use of other energy sources such as fatty acids and amino acids (particularly in resistant disease); protection from antineoplastic agents due to niche interactions with the bone marrow or adipose tissue microenvironments; immune evasion through downregulation of NKG2DL; increased expression of drug efflux pumps; overexpression or increased reliance on antiapoptotic proteins; and upregulation of autophagy in LSCs. In addition, alteration of the LSC gene expression program through hypomethylating agents and other epigenetic modifiers may contribute to LSC-targeting. Created with BioRender.com.

**Figure 2 biomedicines-11-03248-f002:**
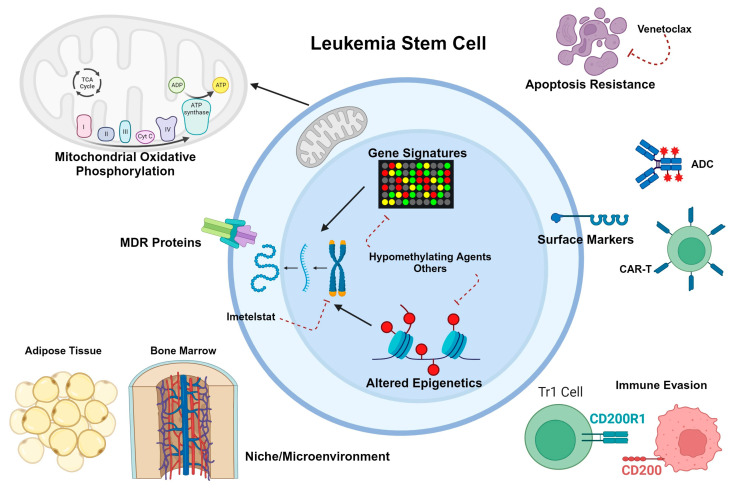
Schematic diagram of what is known about LSC biology in pediatric AML. LSC-selective immunophenotypes, with some but not complete overlap with adult LSCs, can be targeted with antibody-drug conjugates (ADCs) or chimeric antigen receptor T cells (CAR-T). Similar to adult LSCs, pediatric LSC gene expression profiles can be altered through epigenetic modifying agents such as hypomethylating drugs, histone deacetylase inhibitors, and others. Oxidative phosphorylation appears to be similarly crucial for pediatric LSC metabolism as demonstrated in adult LSCs, although little is known about the use of other energy sources. Drug efflux pumps, increased dependence on antiapoptotic proteins, and immune de-activation through CD200 expression have also been reported as survival mechanisms for pediatric LSCs. The bone marrow microenvironment is known to protect LSCs against chemotherapy toxicity, but less is known about the role of adipocytes in therapy resistance. Created with BioRender.com.

**Table 1 biomedicines-11-03248-t001:** Select immunotherapy clinical trials for pediatric patients with AML.

Class/Target	Agent	Study Population	Details	NCT #	Phase	Enrolling?
CD33 ADC	Gemtuzumab ozogamicin	ND secondary or R/R	+ Liposomal Daunorubicin and Cytarabine	NCT04915612	I	Yes
		ND	COG AAML1831; + SOC chemo or CPX-351 + GO	NCT04293562	III	Yes
CD33 CAR T cells	CD33CART	R/R	Lymphodepletion with Fludarabine and Cyclophosphamide	NCT03971799	I/II	Yes
	SC-DARIC33 CAR T cells	R/R	Intermittent oral rapamycin which activates DARIC	NCT05105152	I	Yes
CD123-toxin fusion protein	Tagraxofusp	R/R & CD123+	monotherapy or + Fludarabine/Cytarabine or + Dexamethasone/Vincristine or + Azacitidine	NCT05955261	I	Yes
		R/R	+ Gemtuzumab Ozogamicin	NCT05716009	I	Pending
CD123xCD3 DART	Flotetuzumab	R/R	COG PEPN1812	NCT04158739	I	Completed
CD123/NK cell engager	SAR443579	R/R	Monotherapy; B-ALL and high-risk MDS also eligible	NCT05086315	I/II	Yes
CD123 CAR T cells	CD123-CAR T	R/R & CD123+	Lymphodepletion with Fludarabine and Cyclophosphamide	NCT04318678	I	Active, not recruiting
	CART123	R/R	Lymphodepletion with Fludarabine and Cyclophosphamide	NCT04678336	I	Active, not recruiting
Checkpoint inhibitors	Nivolumab	R/R	+ Azacitidine	NCT03825367	I/II	Active, not recruiting
FOLR1-CDC	ELU001	R/R & CBFA2T3::GLIS2+	Monotherapy	NCT05622591	I	Pending

ADC = antibody-drug conjugate; ND = newly diagnosed; R/R = relapsed/refractory; COG = Children’s Oncology Group; SOC = standard of care; CAR = chimeric antigen receptor; DART = dual-affinity retargeting protein; NK = natural killer; B-ALL = B-cell acute lymphoblastic leukemia; MDS = myelodysplastic syndrome; FOLR1 = folate receptor alpha; CDC = C’Dot drug conjugate.

**Table 2 biomedicines-11-03248-t002:** Non-immunotherapy agents in clinical trials for pediatric patients with AML.

Class/Target	Agent	Study Population	Details	NCT #	Phase	Enrolling?
FLT3 Inhibitors	Gilteritinib	ND	COG AAML1831; + SOC chemo or CPX-351 + GO	NCT04293562	III	Yes
	Gilteritinib	R/R	+ FLAG	NCT04240002	I/II	Yes
	Quizartinib	R/R	+ Fludarabine/Cytarabine/Etoposide	NCT03793478	I/II	Yes
	Pexidartinib	R/R	Monotherapy (National Cancer Institute)	NCT02390752	I	Yes
	MRX-2843	R/R	Monotherapy	NCT04872478	I	Yes
Menin Inhibitors	Revuminib	R/R, KMT2A-r	+ chemo (FLAG for AML)	NCT05761171	II	Pending
		R/R	+ decitabine/cedazuridine (ASTX727) + venetoclax (SAVE trial, MD Anderson)	NCT05360160	I/II	Yes
		R/R, KMT2A-r, NPM1-m	monotherapy	NCT04065399	I/II	Yes
	Ziftomenib	R/R	expanded access on case-by-case basis	NCT05738538	n/a	Available
	JNJ-75276617	R/R, KMT2A-r, NPM1-m, NUP98-r	+ chemo (FLAG for AML), orally bioavailable	NCT05521087	I	Pending
BCL2 Inhibitors	Venetoclax	ND	+ conventional chemo (AML23, St. Jude)	NCT05955261	II	Yes
		R/R	+ FLA + GO; azacitidine (HMA) or ven/aza maintenance if unable to proceed to HSCT (PedAL/EuPAL)	NCT05183035	III	Yes
		R/R	+ idasanutlin (or idasanutlin + chemo)	NCT04029688	I/II	Yes
		R/R	+ azacitidine (HMA) + vorinostat (HDACi) + Fludarabine/Cytarabine/G-CSF	NCT05317403	I	Yes
		R/R	+ Selinexor (XPO1 inhibitor) + FLAG (SELCLAX, St. Jude)	NCT04898894	I	Yes
NF-kB Inhibitor, CREB Inhibitor	Niclosamide	R/R	monotherapy	NCT05188170	I	Yes
E-selectin Antagonist	Uproleselan	ND or 1st R/R	+ HSCT conditioning	NCT05569512	I/II	Yes
PARP Inhibitors	Talazoparib	R/R	POE22-01; + topotecan/gemcitabine	NCT05101551	I	Yes
IDH2 Inhibitor	Enasidenib	R/R	COG ADVL18P1; monotherapy	NCT04203316	II	Yes
NEDD8 Inhibitor	Pevonedistat	R/R	COG/PEP-CTN (ADVL1712); + azacitidine + FLA	NCT03813147	I	Active, not recruiting

ND = newly diagnosed; R/R = relapsed/refractory; COG = Children’s Oncology Group; SOC = standard of care; GO = gemtuzumab ozogamicin; HSCT = hematopoietic stem cell transplant; FLA(G) = fludarabine/cytarabine(/G-CSF); HMA = hypomethylating agent; HDACi = histone deacetylase inhibitor; CREB = cAMP response element binding protein.

## Data Availability

This is a review article, therefore all data supporting reported results are published and cited in the bibliography.

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
