# Peer review of "Emerging and Future Targeted Therapies for Pediatric Acute Myeloid Leukemia: Targeting the Leukemia Stem Cells"

_biomedicines, 2023, doi:10.3390/biomedicines11123248_

Round 1
Reviewer 1 Report
Comments and Suggestions for Authors
The review of Lindsey A. Murphy and Amanda C. Winters “Emerging and future targeted therapies for pediatric acute myeloid leukemia: targeting the leukemia stem cells” is well-organized manuscript elucidating the key points of leukemia stem cells (LSCs) pathophysiology and focusing on LSCs-targeted immunotherapies in pediatric AML. The authors describe both well-studied LSCs-targeted therapies (such as CD33- and CD123-targeted agents, checkpoints inhibitors, etc.) and new promising agents (such as FLT3, menin and PARP inhibitors, epigenetic modifiers and many others).
I have only one comment regarding table design. All tables must be presented in the format of tables, but not as figures. This moment especially applies to Table 2, which is completely unreadable.
Author Response
Thank you for your review. We agree that the Tables did not maintain visibility as we had hoped (particularly Table 2) so we have reviewed the formatting and integrated both tables into the main manuscript in landscape formatting. We hope that this corrects any issues with readability.
Reviewer 2 Report
Comments and Suggestions for Authors
The authors reported a comprehensive review of target therapy in acute myeloid leukemia in children, focusing on leukemia stem cells. This is an attractive topic, on which a great amount of literature has been published, more than half in the last five years. I appreciate a lot the wide scenario of agents reported in the review, which makes the paper suitable also for consultation of more recent study and compounds. This obviously makes the paper more difficult to read, however without affecting the quality of the work. Probably the reference list is too long, corresponding to the extensive literature review.
I would like to suggest
1) to provide a more detailed legend for figures 1 and 2, and 2) to rearrange table 2 which is extremely difficult to read, with large cells and very small font
3) to remove references that are not essential references, limiting to the one mor important (for ex. ref 145-147)
Author Response
Thank you for your review. Below are the responses and corrections made in light of the comments:
- We have added additional text in the figure legends for both Figures 1 and 2 (highlighted).
- We have changed both tables from figures to integrated tables in landscape format which hopefully resolves the issue of readability with Table 2 (both headings highlighted to indicate that the entire table was modified).
- We have carefully reviewed the manuscript and references and removed a small number of non-essential references that were duplications of others (including reference 145). However, we feel that the bulk of the references (where several are listed) add unique information or background that is not replaceable with other references. If there is a reference limit (we did not see one in the author guidelines) then we are happy to look further at the reference list to see what we are able to cull.